# Artificial Intelligence in Colorectal Cancer Surgery: Present and Future Perspectives

**DOI:** 10.3390/cancers14153803

**Published:** 2022-08-04

**Authors:** Giuseppe Quero, Pietro Mascagni, Fiona R. Kolbinger, Claudio Fiorillo, Davide De Sio, Fabio Longo, Carlo Alberto Schena, Vito Laterza, Fausto Rosa, Roberta Menghi, Valerio Papa, Vincenzo Tondolo, Caterina Cina, Marius Distler, Juergen Weitz, Stefanie Speidel, Nicolas Padoy, Sergio Alfieri

**Affiliations:** 1Digestive Surgery Unit, Fondazione Policlinico Universitario A. Gemelli IRCCS, Largo Agostino Gemelli 8, 00168 Rome, Italy; 2Faculty of Medicine, Università Cattolica del Sacro Cuore di Roma, Largo Francesco Vito 1, 00168 Rome, Italy; 3Institute of Image-Guided Surgery, IHU-Strasbourg, 67000 Strasbourg, France; 4Department for Visceral, Thoracic and Vascular Surgery, University Hospital and Faculty of Medicine Carl Gustav Carus, Technische Universität Dresden, 01307 Dresden, Germany; 5National Center for Tumor Diseases (NCT), Partner Site Dresden, 01307 Dresden, Germany; 6ICube, Centre National de la Recherche Scientifique (CNRS), University of Strasbourg, 67000 Strasbourg, France

**Keywords:** artificial intelligence, colorectal cancer, colorectal surgery

## Abstract

**Simple Summary:**

Computer vision (CV) is a field of artificial intelligence (AI) that deals with the automatic analysis of videos and images. Recent advances in AI and CV methods coupled with the growing availability of surgical videos of minimally invasive procedures have led to the development of AI-based algorithms to improve surgical care. Initial proofs of concept have focused on fairly standardized procedures such as laparoscopic cholecystectomy. However, the real value of CV in surgery resides in analyzing and providing assistance in more complex and variable procedures such as colorectal resections. This manuscript provides a brief introduction to AI for surgeons and a comprehensive overview of CV solutions for colorectal cancer surgery.

**Abstract:**

Artificial intelligence (AI) and computer vision (CV) are beginning to impact medicine. While evidence on the clinical value of AI-based solutions for the screening and staging of colorectal cancer (CRC) is mounting, CV and AI applications to enhance the surgical treatment of CRC are still in their early stage. This manuscript introduces key AI concepts to a surgical audience, illustrates fundamental steps to develop CV for surgical applications, and provides a comprehensive overview on the state-of-the-art of AI applications for the treatment of CRC. Notably, studies show that AI can be trained to automatically recognize surgical phases and actions with high accuracy even in complex colorectal procedures such as transanal total mesorectal excision (TaTME). In addition, AI models were trained to interpret fluorescent signals and recognize correct dissection planes during total mesorectal excision (TME), suggesting CV as a potentially valuable tool for intraoperative decision-making and guidance. Finally, AI could have a role in surgical training, providing automatic surgical skills assessment in the operating room. While promising, these proofs of concept require further development, validation in multi-institutional data, and clinical studies to confirm AI as a valuable tool to enhance CRC treatment.

## 1. Introduction

Colorectal cancer (CRC) represents the third most common malignancy in both men and women [1] and the second most common cause of cancer-related deaths worldwide [1,2]. Approximately 60–70% of patients with clinical manifestations of CRC are diagnosed at advanced stages, with liver metastases present in almost 20% of cases [3]. Of importance, 5-year overall survival rates drop from 80–90% in case of local disease to a dismal 10–15% in patients with metastatic spread at the time of diagnosis [4].

While novel diagnostic and pharmacological multimodal approaches significantly improved the quality of CRC screening, diagnosis, and perioperative care [5], several challenges remain to be overcome in the surgical treatment of CRC. Surgery currently represents the gold standard of treatment of CRC, but it is still burdened by a relatively high rate of intraoperative adverse events such as hemorrhage, ureteral lesions, nerve injuries, and lesions to adjacent abdominal organs, and post-operative complications such as anastomotic leakage, ileus, and bleeding [6,7]. Such complications need to be prevented or, at least, promptly recognized and treated to ameliorate patients’ outcomes. 

Today, the vast amount of digital data produced during surgery [8] could be collected and modeled with advanced analytics such as artificial intelligence (AI) to gain insights into root causes of adverse events and to develop solutions with the potential to increase surgical safety and efficiency [9]. 

However, AI applications in surgery are still in their infancy [10] and no AI algorithm has been approved for clinical use to date. So far, AI for intraoperative assistance has mostly focused on minimally invasive surgery given the availability of endoscopic videos natively guiding such procedures. Early proof-of-concept studies have focused on workflow recognition [11], using computer vision (CV) to recognize surgical phases and to detect surgical instruments in endoscopic videos of laparoscopic cholecystectomy [12,13,14,15], inguinal hernia repair [16], and bariatric and endoscopic procedures [17,18]. Due to the availability of data, relative surgical standardization and well-defined clinical problems, AI applications of higher surgical semantics have mostly been focusing on the analysis of laparoscopic cholecystectomy. In fact, recently, AI models were designed and trained to use semantic segmentation to assess correct planes of dissections [12] and define fine-grained hepatocystic anatomy [13], to assess the criteria defining the critical view of safety to prevent bile duct injuries [13], and to automatically provide selective video documentation for postoperative documentation and quality improvement [14,15].

With regard to CRC, AI has shown particularly promising results in screening, with several studies demonstrating the clinical value of AI-assisted colonoscopy for polyp detection and characterization [19,20], and staging, outperforming radiologists in detection of nodal metastases in computed tomography (CT) and magnetic resonance imaging (MRI) images [21].

However, AI applications to enhance the surgical care of CRC have only recently started to appear despite the ubiquity and importance of colorectal surgery. The length, variability, and complexity of colorectal procedures are probably key factors defining both the opportunities and the challenges with applying AI in CRC surgery [22].

The aim of this review is to provide a brief introduction on AI concepts and surgical AI research, and to give an overview of currently available AI applications in colorectal surgery.

## 2. A Brief Introduction to AI for Surgeons

AI is an umbrella term referring to the study of machines that emulate traits generally associated with human intelligence, such as perceiving the environment, deriving logical conclusions from these perceptions, and performing complex actions. AI applications in medicine are steadily increasing, and have already demonstrated clinical impact in various fields including dermatology [23], pathology [24,25], and endoscopy [26,27].

Medical decisions are usually not binary, but highly complex and adaptable with regard to timing (i.e., oncological treatment course, timing of diagnostic procedures), invasiveness (i.e., extent of surgery), and depend on available human and technological resources. In most cases, such choices are made not only on the basis of logical rules and guidelines, but also integrate professional experience. Given the plethora of variation possibilities, it would be extremely complex, if not impossible, to explicitly program machines to perform complex medical tasks, such as understanding free text in electronic health records to stratify patients or interpreting radiological images to make diagnoses. However, the cornerstone of AI is the ability of machines to learn with experience. In machine learning (ML), “experience” corresponds to data. In fact, ML algorithms are designed to iterate over large-scale datasets, identify patterns, and optimize their parameters to better solve a specific problem. While the term strong or general AI relates to the aspiration to create human-like intellectual competences and abstract thinking patterns, currently available AI applications—not only in the field of medicine—are limited to very specific (and in many cases simplified) problems, generally referred to as *weak* or *narrow* AI. In the last two decades, deep learning (DL), a subset of ML, has shown unprecedented performances in the analysis of complex, unstructured data such as free text and images. DL uses multilayer artificial neural networks (ANNs), collections of artificial neurons or perceptrons inspired by biological neural networks, to derive conclusions based on patterns in the input data [28]. In medicine and surgery, a large amount of data is visual, in the form of images (e.g., radiological, histopathological) or videos (e.g., endoscopic and surgical videos). In addition, videos natively guide minimally invasive surgical procedures and could be analyzed for intraoperative assistance and postoperative evaluations. This brief introduction will hence focus on CV, the subfield of AI focusing on machine understanding of visual data [29].

Based on the schematic introduction of key AI-related concepts and terms, the following section will provide a brief overview of a typical surgical AI pipeline in the field of CV (Figure 1). While automated surgical video analysis will be used as an example in the following section, similar approaches can be applied to other types of medical imaging and, in modified structure, to medical data in general.

Once a clinical need has been clearly defined, an appropriate, large-scale, and representative dataset needs to be generated. To verify data appropriateness, it is good practice to see if subject-matter experts (i.e., surgeons) routinely acquire such data and can consistently solve the identified problem using this type of data. For instance, if we want to train a machine to automatically assess the critical view of safety in videos of laparoscopic cholecystectomy, it is important to verify surgeons’ inter-rater agreement in assessing such view and, eventually, devise strategies to formalize and improve such assessments [30]. The inter-rater agreement of experts can also be used to roughly estimate the amount of data necessary to train and test an AI model, as lower inter-rater agreements are generally found in more complex problems that require larger datasets to solve. Finally, since AI performance is heavily dependent on the quality of data used during training, it should be verified that the dataset accurately represents the setting of foreseen clinical deployment. Using the same example of laparoscopic cholecystectomy, acute and chronic cholecystitis cases should be included in the dataset if we want the AI to work in both scenarios.

A further, essential step in generating a dataset for AI is annotation. The term annotation describes the process of labeling data with the information the AI should learn to predict. The type of information to annotate depends on the problem the algorithm is intended to solve. For instance, temporal annotations (e.g., timestamps) are needed to train an AI model to classify surgical steps while spatial annotations (e.g., bounding boxes or segmentations) are required to train an AI model to detect anatomical structures within an image. Regardless of the use case, high-quality annotations are essential for training AI using supervised learning approaches, currently the most common type of learning, as contrasting annotations will significantly hamper training of an AI algorithm. In the context of evaluating the accuracy of an AI algorithm for image recognition, it is important to consider that annotations also serve as “*ground truth*” for comparison. In fact, predictions of the previously trained AI are compared to experts’ annotations to compute performance metrics. The greater the overlap between the annotations and the predictions, the better the algorithm is. Consequently, the reliability of annotations defines the validity of AI assessments. The development and improvement of methods to assess the quality of annotations are subject to ongoing scientific discussion [31,32]. Generally, reporting annotation protocols [33], details on annotators’ expertise, as well as integrating a thorough annotation review process involving multiple annotators and expert reviewers while reporting inter-rater agreements allow to scrutinize annotations.

The annotated dataset should then be split into a training set, used to develop the AI algorithm through multiple iterations, and a test set, used to evaluate the AI performance on unseen data. Split ratio can vary, but it is important to prevent data leaks between training and testing subsets. Of primary importance, test data should not be exposed during training. In addition, testing data should remain as independent as possible from the training dataset. Specifically, this means that not only all image data from one surgical video should be assigned to either the training or the test dataset, but also that serial examinations from one patient (i.e., multiple colonoscopy videos over time) should be treated as a coherent sequence that should not be separated between the training and test dataset. 

At this stage, the dataset and task of interest will be explored to select the best AI architecture or algorithm to then refine, train, and test. In most cases, healthcare professionals and computer scientists collaborate in this process. Interdisciplinary education is, therefore, critical to enable all partners to understand both the clinical and the algorithmic perspectives, to critically appraise related literature, and to overall facilitate a constructive interdisciplinary collaboration. Specifically, involved healthcare professionals should understand and participate in the selection of metrics used to evaluate AI performance [34]. The most commonly used metrics to evaluate how well an AI solves a given task describe the overlap between the true outcome or the annotated “*ground truth*” and the AI prediction. Munir et al. provide a detailed description of commonly used metrics in clinical AI [35]. An important challenge in metric selection is the fact that these overlap metrics are merely surrogate parameters for the clinical benefit. This underlines the need for continuous clinical feedback during the entire process of conceptualization and evaluation of AI applications. Since events to be predicted are often rare (i.e., surgery complications), datasets are commonly unbalanced towards positive or negative cases and require balanced metrics for reliable AI performance assessment. In addition, different clinical applications should optimize different metrics. For instance, screening applications where the cost of a false negative is high, as in computer-aided detection of polyps during screening colonoscopy, should value sensitivity over specificity. In turn, when assessing safety measures such as the critical view in laparoscopic cholecystectomy, the cost of a false positive is high, which is why specificity should be favored over sensitivity. Similar to reporting of annotations, the selected metrics should be transparently reported including specifications about the computing process and underlying assumptions about measured (surrogate) parameters. This is particularly important, as purely technical metrics often fail to predict actual clinical value and ongoing research is looking at developing evaluation methods and metrics specifically for surgical AI applications.

Regardless of how well surgical AIs have been developed and tested, external validation and translational studies are essential to evaluate the clinical potential. Since AI performance is notably dependent on training data, testing on multicentric data reflecting different acquisition modalities, patient populations, and hospital settings is necessary to evaluate how well AI systems generalize outside of the development setting. However, very few external validations studies have been performed to date [15,36] since most open-access datasets only contain data from single centers (Table 1). In such scenarios, multi-institutional collaboration is one of the most influential prerequisites for the development of clinically relevant AI applications. 

To conclude, well designed implementation studies looking at how to integrate such technology in complex clinical and surgical workflows and assessing how these changes impact patient care are crucial to measure actual value for patients and healthcare systems. Translational studies exploring the clinical value of surgical AI still remain to be published, but currently available guidelines can help designing protocols [37], early assessments [38], and reporting of AI-based interventions [39].

## 3. State-of-the-Art of the Intraoperative Application of AI 

As for the majority of surgical procedures, colorectal surgery outcomes are largely based on the efficacy of intraoperative decisions and actions. Accordingly, situational awareness, decision-making, technical skills, and complementary cognitive and procedural aspects are essential for successful surgical procedures. In this context, surgical workflow recognition could improve operating room (OR) staff situational awareness. In addition, the development of AI-based context-aware systems could support anatomy detection, and trigger warnings about dangerous actions, ultimately improving surgeons’ decision-making. Finally, AI applications could be used to enhance technical skills training and performance assessment. 

The following section will provide an overview of the literature on AI for phase and action recognition, intraoperative guidance, and surgical training in colorectal surgery. 

In order to evaluate these potential capabilities, the following evaluation metrics were analyzed by different authors:

Accuracy

Accuracy is defined as the evaluation metric that measures how often the algorithm correctly classifies a data point. More specifically, accuracy determines how many true positive (TP), true negative (TN), false positive (FP), and false negative (FN) results are correctly classified: Accuracy = (TP + TN)/(TP + TN + FN + FP)

TP is the correct classification of a positive class, while TN is the correct classification of a negative class.

FP is the incorrect prediction of the positives, while FN in the incorrect prediction of the negatives.

Sensitivity

It is the rate of positive items correctly identified:Sensitivity = (TP)/(TN + FN)

Specificity

It is the number of negative items correctly identified:Specificity = (TN)/(TN + FN)

Intersection over Union (IoU)

IoU is a method to quantify the similarity between two sample sets:IoU = (TP)/(TP + FN + FP)

F1 Score

F1 score is an evaluation metric that determines the predictive performance of a model by the combination of the precision and recall metrics:F1 = 2 × (Precision × Recall)/(Precision + Recall)

Precision determines the percentage of correct TPs within the predicted ones:Precision = (TP)/(TP + FP)

Recall defines how many TPs the model succeeded to find within all the TPs:Recall = (TP)/(TP + FP)

Dice coefficient

Dice coefficient is a statistical tool that gives the similarity rate between two samples of data:Dice coefficient = (2 × TP)/(2 × TP + FN + FP)

Receiver operating characteristics curve (ROC curve)

The ROC curve is a metric that determines the performance of a classification model at all classification thresholds. The ROC curve plots the true positive rate (TPR) and the false positive rate (FPR):TPR = (TP)/(TP + FN)
FPR = (FP)/(TN + FP)

Area under the curve (AUC)

The AUC gives an aggregate measure of performance across all possible classification thresholds. It measures the entire area under the entire ROC curve and has a range from 0 to 1. A model whose predictions are 100% wrong has an AUC of 0, while a model whose predictions are 100% right has an AUC of 1.

Reviewed studies are summarized in Table 2.

### 3.1. Phase and Action Recognition

Phase recognition is defined as the task of classifying surgical images according to predefined surgical phases. Phases are elements of surgical workflows necessary to successfully complete procedures and are usually defined by consensus and annotated on surgical videos [40]. Accurate AI-based surgical phase recognition has today been demonstrated in laparoscopic cholecystectomy [41,42], gastric bypass surgery [17], sleeve gastrectomy [18], inguinal hernia repair [16], and peroral endoscopic myotomy (POEM) procedures [43].

With regard to current evidence in colorectal surgery, Kitaguchi et al. [44] constructed a large, multicentric dataset including 300 videos of laparoscopic colorectal resections (235 sigmoid resections and 65 high anterior rectal resections) to train and test an AI model on phases and action recognition. Videos were divided into 9 different phases while actions were classified into dissection (defined as the actions for tissue separation), exposure (defined as actions for tissue planes definitions) and other (including all those actions not definable as dissection or exposure, i.e., cleaning the camera, extracorporeal actions, or transection of the colon). Temporal video annotation was performed under the supervision of two colorectal surgeons and led to a dataset containing a total of 82.6 million frames labeled with surgical phases and actions. The authors conducted an out-of-sample validation, consisting in splitting data into test and training sets (16.6 million images from 60 videos and 66 million images from 240 videos, respectively). The generation of the model was based on the training data, while its validation was performed on a set of unseen test data. Overall, the average accuracy for phase classification was 81%, with a pick accuracy of 87% for the transection and anastomosis phases. Performance in action recognition was similarly high, with an accuracy in distinguishing dissection from exposure actions of 83.2%, and a 98.5% accuracy in recognizing the “other” phase. Additionally, the same study also experimented with semantic segmentation of surgical instruments, i.e., identifying pixels of an image depicting a certain instrument. For this purpose, five tools (grasper, point dissector, linear dissector, Maryland dissector, and clipper) were manually segmented on a subset of 4232 images extracted from the above-described dataset. U-Net, a popular AI model for semantic segmentation, was trained on 3404 images and tested on the remaining 20% of the dataset, resulting in mean IoU ranging between 33.6% (grasper) and 68.9% (point dissector).A similar analysis was conducted by the same authors [45] on a dataset of 71 sigmoidectomies annotated with 11 non-sequential phases and surgical actions (dissection, exposure, and other). In addition, extracorporeal (out-of-body images, e.g., camera cleaning) and irrigation (e.g., pelvic and intrarectal irrigation) scenes were also annotated. Phase recognition had an overall accuracy of 90.1% when tested on unmodified phases (Phases 1–9) and 91.9% when phases showing the total mesorectal excision (TME) (phase 1 and 6) and the medial mobilization of colon (phase 2 and 4) were combined. Of importance, they were able to run the phase recognition model at 32 frames per second (fps) on a specialized hardware, to allow real-time phase detection. Finally, the overall accuracy of the extracorporeal action recognition model and the irrigation recognition model were 89.4% and 82.5%, respectively.

Automated phase recognition was also proposed for transanal total mesorectal excision (TaTME) to facilitate intraoperative video analysis for training and dissemination of this technically challenging procedure. 

To develop such an AI model, Kitaguchi et al. [46] retrospectively collected 50 TaTME videos and annotated 5 major surgical phases: (1) purse-string closure; (2) full thickness transection of the rectal wall; (3) down-to-up dissection; (4) dissection after rendezvous; and (5) purse-string suture for single stapling technique; moreover, phases (3) and (4) were further subdivided in four steps: dissection for anterior, posterior, and both bilateral planes. The AI algorithm showed an overall accuracy of 93.2% and 76.7% in phase recognition alone and in combined phase and step recognition, respectively.

### 3.2. Intraoperative Guidance

TME is a complex surgical step most commonly performed within rectal resection for colorectal cancer [47]. This procedure requires dexterity and experience to achieve both oncological radicality and preserve presacral nerves responsible for continence and sexual function [48,49,50]. Given that incomplete TME is directly associated with local tumor recurrence and reduced overall survival, TME represents a crucial step in rectal cancer treatment [51,52,53]. In a recent study [54], supervised machine learning models were used to identify various anatomical structures throughout different phases of robot-assisted rectal resection and to recognize resection planes during TME (Figure 2). In this monocentric study, detections were most reliable for larger anatomical structures such as Gerota’s fascia or the mesocolon with mean F1 scores of 0.78 and 0.71, respectively, the mean F1 score for detection of the dissection plane (“angel’s hair”) during TME was 0.32, while the exact dissection line could be predicted in few images (mean F1 score: 0.05). 

Similarly, Igaki et al. [55] used 32 videos of laparoscopic colorectal resections to train a DL model to automatically recognize the avascular and areolar tissues between the mesorectal fascia and the parietal pelvic fascia for a safe and oncologically correct TME. A total of 600 intraoperative images capturing the TME scenes were extracted and manually annotated by a colorectal surgeon. The AI model was able to recognize the correct oncological dissection plane with a dice coefficient of 0.84. Despite the small sample size of the cohort of this study, potential selection of surgeries with well-identifiable tissue planes, as well as the ambiguous use of the term “TME” in the context of a left-sided hemicolectomy represent limitations, these results show promising preliminary data for automated recognition of the TME plane in the future. Furthermore, these algorithms could also be applied to analysis of surgical skill level, potentially increasing efficiency and improving surgical learning curves.

### 3.3. Indocyanine Green Fluorescence in Colorectal Surgery: Can Artificial Intelligence Take the Next Step?

Indocyanine green (ICG) fluorescence is a reproducible and noninvasive technique employed for bowel perfusion assessment that has become common practice in colorectal surgery during recent years [56,57]. More specifically, ICG is a molecule approved by the Food and Drug Administration (FDA) in 1959 with a vascular-dependent distribution-diffusion and its fluorescence could be detected by a near infrared camera [58,59]. Although several meta-analyses have demonstrated that ICG angiography leads to a reduction of anastomotic leaks [60,61,62], two randomized controlled trials (RCTs) did not indicate significant advantages as compared to perfusion assessment under standard white light [63,64]. Several authors have proposed a quantitative analysis through ICG curves to address the lack of standardization into the interpretation of ICG angiography images and data [57]. However, the different anatomy of patients’ microcirculation results in a high interindividual variability of ICG quantitative parameters, hindering the objective interpretation of the visualization. Intraoperative application of AI for analysis of ICG fluorescence could greatly enhance the identification of colonic malperfusion and could overcome the limits of quantitative analysis. A recent study [65] highlights that the real-time microcirculation analysis using AI algorithms is achievable and results in significantly higher accuracy compared to the conventional method, namely T1/2 max (time from first fluorescence increase to half of maximum), TR (time ratio: T1/2 max/Tmax, where Tmax is defined as the time form first fluorescence increase to maximum), and RS (rise slope) [38]. 

Park et al. [65] employed unsupervised learning, an AI training paradigm not requiring annotations during training, to cluster 10,000 ICG curves into 25 patterns using a neural network which mimics the visual cortex. The ICG curves were extracted from 65 procedures. Curves were preprocessed to decrease the degradation of the AI model due to external influences such as light source reflection, background and camera movements. From the comparison of the AUC of the AI-based method with T1/2 max, TR and RS, the AI model demonstrated a higher accuracy in the microcirculation evaluation, with values of 0.842, 0.750, 0.734, and 0.677, respectively. This facilitates creation of a color mapping system of red-green-blue areas classifying the grade of vascularization. 

Although based on a limited number of images, this AI model provides a more objective and accurate method of fluorescence signal evaluation as compared to purely visual examination by a surgeon. Moreover, the real-time analysis may give an immediate evaluation of the grade of perfusion during the minimally invasive colorectal procedures for a prompt recognition of insufficient vascularization. 

### 3.4. AI for Surgical Training in Colorectal Surgery

Surgical training is another area where surgical AI could have a great impact. Over the past decade, different reproducible systems to manually assess technical skills during open and laparoscopic procedures have been established [66,67,68]. Recent works have analyzed the potential of AI to automate such systems for technical skill analysis. For instance, Azari et al. [69] demonstrated that a computer algorithm could predict expert surgeons’ skill ratings during surgical tasks such as knot tying or suture in a total of 103 video clips from 16 laparoscopic operations. Three expert surgeons observed and rated each clip for motion economy, fluidity of motion, and tissue handling during suturing or tying tasks with values from 0 to 10. A custom video tracking software was used to trace a region of interest (ROI) around the surgeon’s hand. The measurements in each clip of surgeon’s hand displacement, speed, and acceleration along with jerk and spatiotemporal curvature were implemented to assess surgical dexterity. The authors, finally, applied a set of linear regression models to use the kinematic features of each clip to predict the expert ratings. In fact, they assumed that motion economy could be assessed by measuring the surgeon’s hand speed in a consistent area, while fluidity could correspond to a low number of changes in speed during the action. Their results show that an AI model can appropriately assess fluidity of motion and motion economy during the suturing tasks, while some limitations exist in assessing more complex actions such as tissue handling. 

Kitaguchi et al. employed AI to assess surgical skill specifically during colorectal procedures. The authors randomly extracted 74 videos from the dataset of the Japan Society for Endoscopic Surgery (JSES) [70]. Starting from 2004, those videos had been examined by 2 or 3 judges following the criteria of the Endoscopic Surgical Skill Qualification System (ESSQS) to assess surgical skills of the operating surgeons. As a whole, 1480 video clips were extracted for each surgical step analyzed, namely medial mobilization, lateral mobilization, inferior mesenteric artery (IMA) transection, and mesorectal transection. Clips were split into a training (80%) and a test (20%) set. AI-based automatic surgical skill classification showed a pick accuracy of 83.8% during IMA transection, while the skill classification during medial and lateral mobilization and mesorectal transection showed an accuracy of 73.0%, 74.3%, and 68.9%, respectively. 

**Table 2 cancers-14-03803-t002:** Schematic summary of the reviewed studies using AI to analyze colorectal procedures.

First Author	Year	Task	Study Design (Study Period)	Cohort	AI Model	Validation	Performance
Kitaguchi, D. [44]	2020	Phase recognition, action classification and tool segmentation	Multicentic retrospective study (2009–2019)	300 procedures (235 LSs; 65 LRRs)	Xception, U-Net	Out-of-sample	Phase recognition mean accuracy: 81.0% Action classification mean accuracy: 83.2% Tool segmentation mean IoU: 51.2%
Park, S.H. [65]	2020	Perfusion assessment	Monocentric study (2018–2019)	65 LRRs	-	Out-of-sample	AUC: 0.842 Recall: 100% F1 score: 75%
Kitaguchi, D. [45]	2020	Phase recognition and action detection	Monocentric retrospective study (2009–2017)	71 LSs	Inception-ResNet-v2	Out-of-sample	Phase recognition (Phases 1–9): - Overall accuracy: 90.1% - Mean Precision: 90% - Mean Recall: 89% - Mean F1-score: 89% Phase recognition (Phases I–VII): - Overall accuracy: 91.9% - Mean Precision: 91% - Mean Recall: 89% - Mean F1-score: 90% Extracorporal action detection: - Overall accuracy: 89.4% - Precision: 96% - Recall: 83% - F1-score: 89% Irrigation action detection: - Overall accuracy: 82.5% - Precision: 96% - Recall: 68% - F1-score: 80%
Kitaguchi, D. [70]	2021	Surgical skill assessment	Monocentric retrospective study (2016–2017	74 procedures (LSs and LHARs)	Inception-v1 I3D	Leave-one-out cross validation	Classification in 3 performance groups, mean accuracy: - Overall: 75.0% - Medial mobilization: 73.0% - Lateral mobilization: 74.3% - IMA transection: 83.8% - Mesorectal transection: 68.9% Predictions of lower performance groups: - Sensitivity: 94.1% - Specificity: 96.5% - AUROC: 0.989 Predictions of upper performance groups: - Sensitivity: 87.1% - Specificity: 86.0% - AUROC: 0.934
Kitaguchi, D. [46]	2022	Phase and step recognition	Monocentric retrospective study (2018–2019)	50 TaTMEs	Xception	Out-of-sample	Phase recognition: - Overall accuracy: 93.2% - Mean Precision: 94% - Mean Recall: 86% - Mean F1-score: 90% Phase and step recognition: - Overall accuracy:76.7% - Mean Precision: 75% - Mean Recall: 76% - Mean F1-score: 75%
Igaki, T. [55]	2022	Plane of dissection recognition	Monocentric study (2015–2019)	32 LSs/LRRs	-	Out-of-sample validation	Accuracy of areolar tissue segmentation: 84%
Kolbinger, F.R. [54]	2022	Phase and step recognition, segmentation of anatomical structures and planes of dissection	Monocentric retrospective study (2019–2021)	57 robot-assisted rectal resections	Phase recognition: LSTM, ResNet50 Segmentation: Detectron2, ResNet50	Phase recognition: 4-fold cross validation Segmentation: Leave-one-out cross validation	Phase recognition: - Mean accuracy: 83% - Mean F1-score: 79% Segmentation of anatomical structures and planes of dissection (selection): Gerota’s fascia: - Mean F1-score: 78% - Mean IoU: 74% - Mean Precision: 80% - Mean Recall: 81% - Mean Specificity: 83% Mesocolon: - Mean F1-score: 71% - Mean IoU: 65% - Mean Precision: 73% - Mean Recall: 74% - Mean Specificity: 77% Mesorectum: - Mean F1-score: 48% - Mean IoU: 45% - Mean Precision: 50% - Mean Recall: 50% - Mean Specificity: 53% Dissection Plane: - Mean F1-score: 32% - Mean IoU: 28% - Mean Precision: 34% - Mean Recall: 35% - Mean Specificity: 39% Dissection Line: - Mean F1-score: 5% - Mean IoU: 4% - Mean Precision: 8% - Mean Recall: 6% - Mean Specificity: 12%

AR: action recognition; AUC: area under the curve; AUROC: area under the receiver operating characteristic; IMA: inferior mesenteric artery; IoU: intersection-over-union; LHAR: laparoscopic high anterior resection; LRR: laparoscopic rectal resection; LS: laparoscopic sigmoidectomy; Ta-TME: transanal total mesorectal excision.

## 4. Discussion

The recent application of AI to the medical field is gradually revolutionizing the diagnostic and therapeutic approach to several diseases. While in some cancer entities such as lung and breast cancer, various AI applications have been implemented and studied, the use of AI in CRC is still in a preliminary phase [71]. In CRC, the utility of AI has primarily been established for assisting in screening and staging. Meanwhile, evidence on colorectal surgery-specific use of AI, i.e., in an intraoperative setting, is scarce. 

After introducing key AI terms and concepts to a surgical audience, this overview summarizes the state-of-the-art of AI in CRC surgery. Included studies develop deep neural networks to recognize surgical phases and actions, provide intraoperative guidance, and automatize skill assessment by analyzing surgical videos. Particularly promising results have been demonstrated for surgical phases and actions recognition, with studies reporting accuracies above 80% [44,45,46]. Intraoperative guidance has been addressed either by using AI to identify safe planes of dissections, as demonstrated by models recognizing the correct TME plane with a dice coefficient of 0.84 [55], and by precisely interpreting fluorescent signals for perfusion assessment [65]. Finally, in 2021, Kitaguchi et al. [70] showed that AI can reliably assess surgical technical skills according to the criteria of the Endoscopic Surgical Skill Qualification System (ESSQS) developed by the Japan Society for Endoscopic Surgery (JSES). While promising, all except one of these studies train and test AI models on data coming from a single center, meaning that the generalizability of performance across centers is still unknown. In addition, annotations protocols are rarely well disclosed and used datasets tend to be private, preventing from scrutinizing the validity of “*ground-truth*” annotations and the representativity of data. Finally, currently, no studies have reported relevant healthcare outcomes, either related to improve surgical efficiency (i.e., decreasing times and costs) or clinical outcomes such as complications or survival rates. Overall, while this review purposely focuses on CRC surgery, these findings align well with those of recent reviews on surgical data science and CV analysis of surgical videos for various applications [10,29,72,73,74]. 

AI applications in CRC screening and staging are more mature. With regard to AI application in CRC screening, multiple studies have demonstrated advantages as compared to routine endoscopic and radiological exams [19,20,21]. As such, a recent meta-analysis of RCTs [75] specifically analyzed the capability of AI algorithms to recognize polypoid lesions in comparison to physicians’ assessment alone. The authors showed a significant increase in the detection rate of polyps when AI was employed together with colonoscopy as compared to colonoscopy alone (OR 1.75; 95% CI: 1.56–1.96; *p* < 0.001). This study also identified an increased identification rate of adenomas when physicians’ assessment of colonoscopy was combined with AI as compared to physicians’ assessment alone (OR1.53; 95% CI: 1.32–1.77; *p* < 0.001), with an absolute improvement in adenoma detection rate ranging from 6% to 15.2%.

Evident benefits have also been demonstrated when AI was employed as a tool for CRC staging, assisting imaging techniques commonly used in clinical practice such as CT scan and MRI. A recent study by Ichimasa et al. [21] reported an AI model assessing lymph node status in 45 patients with T1 CRC by analyzing 45 clinicopathological factors. Herein, sensitivity, specificity, and accuracy were higher in comparison to current guidelines (100%, 66%, and 69%, respectively). Other AI models have shown promising results in assessing and predicting pelvic lymph node status before surgery [76,77] and liver metastasis [78]. Even if the true clinical impact of such systems remains to be investigated in RCTs, they have the potential to enhance appropriate disease staging and consequent treatment allocation.

Similarly, several authors based their research on the AI application to the pathological staging of CRC. In this context, ML models have been developed with the aim of guaranteeing an accurate tumor classification and distinction between malignant and normal colon samples. Rathore et al. [79] created a CRC detection system able to determine malignancy and grading with an accuracy of 95.4% and 93.5%, respectively. Similar results have been reported by Xu et al. [80] who constructed two AI models distinguishing malignant lesions from normal colon samples based on hematoxylin and eosin-stained images, demonstrating an accuracy of 98% and 95%, respectively. 

As compared to AI in screening endoscopy, radiology, and pathology, surgical data and applications are more challenging for several reasons. Surgical videos are dynamic in nature, showing complex-to-model tool–tissue interactions deforming or completely reshaping anatomical instances. In addition, surgical workflows and practices are hard to standardize, especially in long and widely variable procedures such as colorectal surgeries. Finally, surgeons integrate prior knowledge, such preoperative imaging, as well as their experience and intuition to take decisions in the OR. To tackle these challenges, more and better data are essential. This means achieving consensus around annotations protocols [33] and publicly releasing large, high-quality annotated datasets. In this context, the collaboration of multiple institutions is necessary to guarantee that data are diverse and representative [73]. Such datasets will not only be important to train better AI models, but also to prove generalizability through external validation studies [15]. In addition, AI models analyzing multimodal data and capable of causal, probabilistic reasoning will have to be developed if we truly want to imitate or enhance surgeons’ mental models. 

Current AI algorithms mostly focus on individual aspects of medical care instead of truly understanding and mimicking human cognitive behavior. In surgery, complex procedures such as colorectal resections do not represent linear processes, but rather a prudent organization of interconnected information, derived from a combination of multiple features such as surgeons’ skills and experience, patients’ characteristics, and environmental factors [81]. To understand how expert surgeons would perform a specific procedure and the consequent analysis of these qualitative data and cognitive tasks could potentially generate AI algorithms able to replicate experts’ behaviors.

The more data, the better the AI models, the more applications of AI and, specifically, CV, are conceivable for surgical purposes in the near future. For instance, DL-based image processing may be applied for automatic recognition of critical anatomical structures such as vessels, organs, and surgical instruments. As such, real-time CV models could support the surgeon in the correct recognition of surgical planes, essential for an oncologically correct treatment. This could potentially increase surgical safety and efficiency, support the surgeon in decision-making, and potentially reduce the rate of intraoperative adverse events. Furthermore, context-aware CV has the potential advantage of recognizing and classifying errors, giving the opportunity to enhance surgical care both during surgical procedures and in the postoperative course. Since a high number of “near misses” has been described to be associated with a poorer postoperative outcome in CRC treatment [82], AI-based identification of pitfalls of a certain procedure could therefore potentially reduce perioperative complications and share expertise. Such insights at scale hold the potential to even revolutionize the idea of surgical training and coaching, with consistent impacts for both practicing surgeons and trainees. Once reliable quantitative metrics are validated, CV systems will be able to give formative feedback based on specific features such as time spent for the surgical procedure, anatomic structures recognition, and capability to achieve specific surgical tasks. 

## 5. Conclusions

In conclusion, the application of AI in colorectal surgery is currently in an early stage of development, while particularly in screening and diagnosis, AI has already demonstrated clear clinical benefits. In the context of intraoperative AI applications, many preclinical studies, even if with promising results, do not progress to clinical translation, so that the clinical benefit cannot be objectively quantified. The creation of larger datasets, the standardization of surgical workflows and the future development of AI models able to mimic expert surgeons’ abilities could pave the way to the widespread use of AI assistance even in complex surgical procedures such as oncological colorectal surgery.

## Figures and Tables

**Figure 1 cancers-14-03803-f001:**
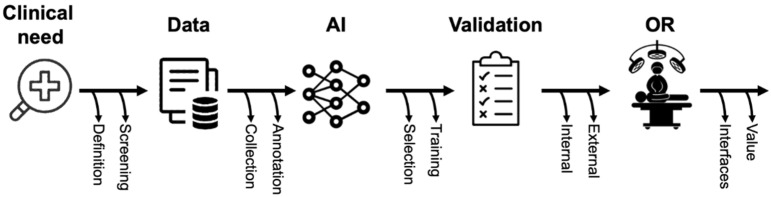
Schematic representation of the phases of surgical AI research.

**Figure 2 cancers-14-03803-f002:**
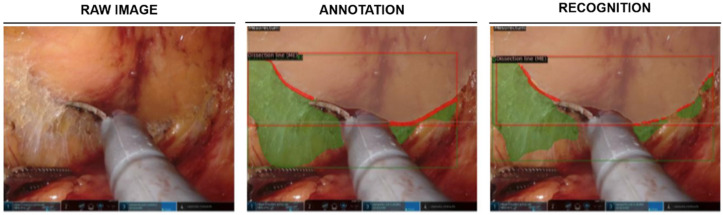
Machine learning-based identification of anatomical structures and dissection planes during TME. Example image displays the mesorectum (light brown), dissection plane (green), and dissection line (red). Figure modified from [53].

**Table 1 cancers-14-03803-t001:** Publicly available annotated datasets of colorectal surgery procedures. The dataset coming from Heidelberg University Hospital has grown in size and annotation types over the years and editions of Endoscopic Vision (EndoVis) challenge, a popular medical computer vision challenge organized at the International Conference on Medical Image Computing and Computer Assisted Intervention (MICCAI).

Name	Year	Procedure (Data Type)	Online Links	Annotation	Size
EndoVis-Instrument	2015	Laparoscopic colorectal procedures *	https://endovissub-instrument.grand-challenge.org/Data/ Accessed on 22 July 2022	Instrument segmentation, center coordinates, 2D pose	270 images, 6 1-min long videos
EndoVis-Workflow	2017	Laparoscopic rectal resection, sigmoidectomy, proctocolectomy (videos, device signals)	https://endovissub2017-workflow.grand-challenge.org/Data/ Accessed on 19 July 2022	Phases, instrument types	30 full-length videos
EndoVis-ROBUST-MIS	2019	Laparoscopic rectal resection, sigmoidectomy, proctocolectomy (videos)	https://www.sciencedirect.com/science/article/pii/S136184152030284X Accessedon 23 July 2022	Instrument types and segmentation	10,040 images, 30 full-length videos
Heidelberg colorectal data	2021	Laparoscopic rectal resection, sigmoidectomy, proctocolectomy (videos, device signals)	https://www.nature.com/articles/s41597-021-00882-2 Accessed on 22 July 2022	Phases, instrument types and segmentation	10,040 images, 30 full-length videos

* The dataset does not specify the type of surgery and also contains videos of robotic minimally invasive surgery on ex vivo porcine organs.

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
