# Peer review of "Artificial Intelligence in Colorectal Cancer Surgery: Present and Future Perspectives"

_cancers, 2022, doi:10.3390/cancers14153803_

Round 1

Reviewer 1 Report

The authors reviewed the state-of-the-art AI applications in colorectal cancer surgery, including phase and action recognition, intraoperative guidance, indocyanine green fluorescence in colorectal surgery, and surgical training in colorectal surgery. To that end, they systematically summarized the literature on existing intraoperative AI applications for surgical purposes and focused mainly on the methods, applications, and performance.

Some revisions should be made: 

(1) Enhance the abstract by providing the significance of the findings.

(2) The authors should add detailed evaluation metrics and the public datasets descriptions with links. See Munir, Khushboo, et al. "Cancer diagnosis using deep learning: a bibliographic review." Cancers 11.9 (2019): 1235. (https://www.mdpi.com/2072-6694/11/9/1235)

(3) The discussion section should be enhanced with regard to future directions, and more comprehensive analysis should be presented.

(4) Please run a careful language edit and spell-check.

Minor comments: line 208 should be State-of-the-art.

Minor comments: line 457, need a reference for this paragraph.

Reviewer 2 Report

To the Authors:

This is a well-written and interesting review of the potential for AI in surgery for colorectal cancer. However, there are some conceptual holes that could be further elaborated:

1) The specific outcomes that will be improved are not clearly stated. The authors discuss phase recognition, intraoperative guidance, enhancement of existing technology (indocyanine green fluorescence), and surgical training as potential targets for AI to enhance. However, these are not outcomes in and of themselves; they are more about improving the process. The goal of AI is to improve overall value by decreasing time and money spent (efficiency) while improving outcomes (not clearly defined). I would suggest a re-frame to focus on the specific outcomes and then fold in these concepts as part of the outcomes. 

2) The paper struggles with talking about universal targets in surgery versus specific problems in colorectal surgery. The authors cite a limited set of colorectal papers, which are helpful. However, it would also be helpful to talk about the evidence generated in the overall surgical landscape and then say why AI has a particular opportunity in colorectal cancer.

3) The surgical focus of the review completely neglects the fact that information gathered prior to the operation is essential in informing the surgeon regarding specific decisions. The world is multimodal, including pre-surgical imaging (radiological or endoscopic), tabular data from the electronic health record, and genomic data. I would include a section discussing this.

Round 2

Reviewer 2 Report

The authors have adequately addressed my concerns.